# A Radiomics Approach on Chest CT Distinguishes Primary Lung Cancer from Solitary Lung Metastasis in Colorectal Cancer Patients

**DOI:** 10.3390/jpm12111859

**Published:** 2022-11-07

**Authors:** Jong Eun Lee, Luu Ngoc Do, Won Gi Jeong, Hyo Jae Lee, Kum Ju Chae, Yun Hyeon Kim, Ilwoo Park

**Affiliations:** 1Department of Radiology, Chonnam National University Hospital, Chonnam National University Medical School, Gwangju, Korea; 2Department of Radiology, Chonnam National University, Gwangju, Korea; 3Department of Radiology, Chonnam National University Hwasun Hospital, Chonnam National University Medical School, Hwasun, Korea; 4Department of Radiology, Research Institute of Clinical Medicine of Jeonbuk National University-Biomedical Research Institute of Jeonbuk National University Hospital, Jeonju, Korea; 5Department of Artificial Intelligence Convergence, Chonnam National University, Gwangju, Korea; 6Department of Data Science, Chonnam National University, Gwangju, Korea

**Keywords:** solitary pulmonary nodule, colorectal neoplasms, lung neoplasms, machine learning, computed tomography

## Abstract

Purpose: This study utilized a radiomics approach combined with a machine learning algorithm to distinguish primary lung cancer (LC) from solitary lung metastasis (LM) in colorectal cancer (CRC) patients with a solitary pulmonary nodule (SPN). Materials and Methods: In a retrospective study, 239 patients who underwent chest computerized tomography (CT) at three different institutions between 2011 and 2019 and were diagnosed as primary LC or solitary LM were included. The data from the first institution were divided into training and internal testing datasets. The data from the second and third institutions were used as an external testing dataset. Radiomic features were extracted from the intra and perinodular regions of interest (ROI). After a feature selection process, Support vector machine (SVM) was used to train models for classifying between LC and LM. The performances of the SVM classifiers were evaluated with both the internal and external testing datasets. The performances of the model were compared to those of two radiologists who reviewed the CT images of the testing datasets for the binary prediction of LC versus LM. Results: The SVM classifier trained with the radiomic features from the intranodular ROI and achieved the sensitivity/specificity of 0.545/0.828 in the internal test dataset, and 0.833/0.964 in the external test dataset, respectively. The SVM classifier trained with the combined radiomic features from the intra- and perinodular ROIs achieved the sensitivity/specificity of 0.545/0.966 in the internal test dataset, and 0.833/1.000 in the external test data set, respectively. Two radiologists demonstrated the sensitivity/specificity of 0.545/0.966 and 0.636/0.828 in the internal test dataset, and 0.917/0.929 and 0.833/0.929 in the external test dataset, which were comparable to the performance of the model trained with the combined radiomics features. Conclusion: Our results suggested that the machine learning classifiers trained using radiomics features of SPN in CRC patients can be used to distinguish the primary LC and the solitary LM with a similar level of performance to radiologists.

## 1. Introduction

Solitary pulmonary nodule (SPN) is often encountered in patients with colorectal cancer (CRC) surveillance. In a previous study of pulmonary nodules in CRC patients, 27% of all patients had pulmonary nodules, of which 56% were SPN [1]. Although it is of clinical importance to render differential diagnosis of SPN, primary lung cancer (LC) can be mistaken for solitary lung metastasis (LM) [2,3]. The differentiation between primary LC and solitary LM is critical for establishing a treatment strategy as well as predicting prognosis in clinical practice. Primary LC requires a radical surgical resection, which shows a relatively good prognosis, while solitary LM requires a minimally invasive surgical resection, but has a relatively poor prognosis [4,5,6]. The clinical standard for the differential diagnosis between primary LC and solitary LM is a histopathological confirmation; however, the histopathologic differentiation requires invasive procedures such as a needle or surgical biopsy. These procedures can pose a potential risk, especially in the lesions that are small and located at deep parenchyma.

Although previous studies have shown that the conventional analysis of computerized tomography (CT) images by radiologist can render a non-invasive diagnosis of pulmonary nodules [7,8], this approach relies on a relatively subjective measurement and its accuracy largely depends on the experience of readers. Recently, there has been an increasing interest in the use of radiomics for the quantitative interpretation of medical imaging data [9] and a machine learning approach for building prediction or classification models in a wide range of medical applications [10,11,12,13]. For example, Beig et al. demonstrated that the radiomics features of lung nodules on CT images can be used for distinguishing between benign and malignant nodules [14]. Radiomics has been shown to reduce inter-reader variability and improve diagnostic efficiency for the quantitative analysis of pulmonary nodules [15].

To date, there have been no studies that have quantitatively analyzed SPN in patients with CRC using a radiomics approach and evaluated its feasibility through comparison with radiologists. The aim of this study was to investigate the feasibility of using machine learning algorithms combined with radiomics analysis in distinguishing primary LC from solitary LM in CRC patients with a SPN. 

## 2. Materials and Methods

### 2.1. Patient Selection

We retrospectively enrolled patients from three different tertiary medical centers: Chonnam National University Hwasun Hospital, Jeollanam-do, Republic of Korea; Chonnam National University Hospital, Gwangju, Republic of Korea; Jeonbuk National University Hospital, Jeollabuk-do, Republic of Korea. A total of 282 patients were enrolled with the following inclusion criteria: (1) CRC patients who had a SPN which measured to be less than 30 mm, (2) a confirmed pathology with either a primary LC or a solitary LM by thoracoscopic wedge resection, and (3) the presence of a preoperative chest CT exam within 2 weeks of histopathologic diagnosis. The following exclusion criteria were applied: (1) the previous history of therapy or biopsy (*n* = 10), (2) SPN that was deemed too small to be characterized at CT (≤6 mm) (*n* = 12), and (3) the absence of contrast-enhanced CT images (*n* = 21). Finally, a total of 239 CRC patients were included in this study. The training and internal testing datasets consisted of 199 patients from Chonnam National University Hwasun Hospital. The external testing dataset consisted of 20 patients from Chonnam National University Hospital and 20 patients from Jeonbuk National University Hospital (Figure 1).

### 2.2. CT Data Acquisition

All CT datasets from the three hospitals were acquired with an in-plane dimension of 512 × 512 voxels; however, the acquisition and reconstruction parameters were slightly different between the datasets.

The chest CT scans of the training and internal testing dataset were obtained using a multi-detector CT scanner, LightSpeed 16 (*n* = 69; GE Healthcare), LightSpeed VCT (*n* = 86; GE Healthcare), Somatom Definition Flash (*n* = 33; Siemens Healthineers), or Revolution (*n* = 11; GE Healthcare), with the following parameters: reconstruction thickness, 2.5–3.0 mm; rotation time, 0.5 to 0.8 s; peak kilovoltage, 120 kVp; tube current, 60–220 mAs; automatic exposure control, reconstruction kernel, and sharp algorithm. Contrast-enhanced chest CT images were obtained after an intravenous injection of 120 to 130 mL nonionic contrast medium, either iohexol (Omnipaque^®®^, GE Healthcare) or iopromide (Ultravist 300^®®^, Bayer AG), at an average injection rate of 2 mL/s.

The chest CT scans of the external testing dataset were obtained using a multi-detector CT scanner, Somatom Definition AS (*n* = 30; Siemens Healthineers) or Philips IQon Spectral CT (*n* = 10; Philips Healthcare), with the following parameters: reconstruction thickness, 2 mm; rotation time, 0.5 s; peak kilovoltage, 120 kVp; tube current, 200 effective mAs; reconstruction kernel and sharp algorithm. Contrast-enhanced chest CT images were obtained after an intravenous injection of 100 to 130 mL nonionic contrast medium, iohexol (Omnipaque^®®^, GE Healthcare), at an average injection rate of 2–2.5 mL/s.

### 2.3. Visual Analysis of CT Images

CT images were interpreted by two board-certified thoracic radiologists (a senior radiologist with more than 10 years of experience and a junior radiologist with 5 years of experience) who were blinded to the clinical and histopathologic information of patients. If their interpretations differed, a decision was made based on a consensus reading of the two thoracic radiologists. If a consensus was not achieved, the senior reader’s interpretation was accepted. 

Semantic CT features such as margin (smooth, lobulated, or spiculated), density (solid or sub-solid), the presence of an air-bronchogram, cavitation, or pleural tags were assessed using the chest CT images obtained with a lung window setting (window width, 1500 HU; level, −700 HU).

### 2.4. Region of Interest (ROI) Segmentation and Radiomics Features Extraction

Intranodular ROIs were semi-automatically segmented in the contrast-enhanced axial CT images by a thoracic radiologist who was blinded to pathologic diagnosis using an open-source software, 3D Slicer (version 4.11, http://www.slicer.org, accessed on 26 February 2021). After the intranodular segmentation was drawn, a perinodular segmentation representing a region outside the nodule up to a radial distance of 15 mm was automatically obtained by dilating the delineated intramodular ROI contours using a built-in function in 3D-slicer (Figure 2).

Radiomics features were extracted from both the intranodular and perinodular ROIs using a PyRadiomics (version 3.0) module in 3D Slicer. A total of 107 original features based on shape (*n* = 14), first-order features (*n* = 18), and texture features (*n* = 75) were computed.

### 2.5. Feature Selection and Model Training

The feature selection process was divided into two steps. First, we applied a linear discriminant analysis to select a subset of features that can preserve the class discriminatory power as much as possible. Second, a regression-based Least Absolute Shrinkage and Selection Operator (LASSO) analysis was applied to select the features that would be used later to train the classifier models. This two-step feature selection process can not only reduce training time, but also strengthen the regularization process.

The linear discriminant analysis was applied using Fisher Discriminant Ratio (FDR) to measure the discriminatory power of individual features between two classes in the first feature selection step. FDR is defined as,
(1)$$FDR=\frac{{\left(m_{1}-m_{2}\right)}^{2}}{\sigma_{1}^{2}+\sigma_{2}^{2}}$$
where $m_{1}$ and $m_{2}$ are the respective mean values, $\sigma_{1}^{2}$ and $\sigma_{2}^{2}$ are respective variances associated with the values of a feature in two classes. All radiomics features were ranked from low to high FDR values and top 30% FDR ranked features were selected. These top 30% features were then used for the LASSO analysis in the second step. Top 10 features with the highest absolute coefficient values from LASSO were selected as the radiomics feature candidates for discriminating primary LC from solitary LM. The process of selecting ten features by LASSO was repeated 100 times. Note that we did not use the held-out internal testing dataset at this step. The training dataset were randomly divided into LASSO-training (70%) and LASSO-validation (30%) sets for each repetition. Every iteration provided a different combination of top 10 feature candidates. Features that were selected as the top 10 features more than 60 times out of 100 repetitions were considered to be robust and reliable and, therefore, used as final features for the classification between primary LC and solitary LM. Appendix A shows the features that satisfied the above-mentioned conditions and were used to train the classifier. Support vector machine (SVM) with a Radial Basis Function kernel [16] was used as a final classifier. SVM can classify the data into two classes based on the hyperplane that has the greatest distance to the support vectors. Similar to the feature selection process, a bootstrapping strategy which repeats the process 100 times with different data partitions was adapted to evaluate the performance of SVM classification. The regularization parameters of SVM were optimized based on validation data performance. The overall workflow of the radiomic model development is displayed in Figure 2.

To assess a segmentation reproducibility, the ROIs were independently drawn by two board-certified thoracic radiologists and intraclass correlation coefficients (ICCs) were assessed. In addition, the interobserver reproducibility of the extracted radiomic features was evaluated by calculating the ICCs of the features that were selected to be used for training the SVM classifiers (Appendix A).

### 2.6. Model Performance Evaluation and Reader Test

The performance of the SVM classifiers for distinguishing primary LC from solitary LM was evaluated using the internal and external testing datasets. In addition, two board-certified thoracic radiologists independently reviewed the internal and external test datasets. They were completely blinded to the clinical information and asked to decide whether each nodule was either primary LC or solitary LM using a binary scale. Their performance, which was accessed using accuracy, sensitivity (primary LC considered as a positive condition) and specificity, was compared to those of the radiomic SVM classifiers.

### 2.7. Comparison between the Radiomics and Semantic CT Imaging Features

To understand the meaning of radiomics features and elucidate their association with semantic CT imaging features, four radiomics features that were selected for the SVM classifiers were compared with the semantic CT features that were statistically different between primary LC and solitary LM. Their association was evaluated by Pearson product-moment correlation coefficient.

### 2.8. Statistical Analysis

The comparisons of the visual CT imaging features between primary LC and solitary LM were performed using a Pearson Chi-square test for the categorical variables and an independent t-test for the continuous variables. 

To evaluate the radiomics model performance for distinguishing primary LC from solitary LM, the accuracy (with a threshold of 0.5), sensitivity, specificity, and the area under the receiver operating characteristic curves (AUC) of the SVM models were calculated for the internal and external testing datasets and compared to those of the two readers. The correlation between the selected radiomics and visual CT imaging features were assessed by Pearson correlation coefficients.

All statistical analyses were performed using SPSS software, version 25.0 (IBM, Armonk, NY, USA). All statistical tests were two-sided, and the *p* values < 0.05 were considered statistically significant.

## 3. Results

The clinical characteristics of patients in the training, internal, and external testing datasets are summarized in Table 1. In the training set, 101 patients were diagnosed with a solitary LM and 58 patients with a primary LC. In the internal test set, 29 patients were diagnosed with solitary LM, 11 patients with primary LC. In the external test set, 28 patients were diagnosed with solitary LM and 12 patients with primary LC. There was no significant difference regarding patient characteristics between either the training and internal test sets or the training and external test sets.

### 3.1. Analysis of Semantic CT Imaging Features

The analysis of semantic CT imaging features in the entire patients are shown in Table 2. The proportion of nodules with spiculated margin (*p* < 0.001), and subsolid density (*p* < 0.001) of the primary LC were significantly higher than those of the solitary LM. The primary LC demonstrated significantly higher frequencies of air-bronchogram (*p* < 0.001) and pleural tags (*p* < 0.001) than the solitary LM. The presence of cavitation was similar between the two groups. The detailed comparison of semantic CT imaging features between training, internal, and external testing datasets is summarized in Appendix A.

### 3.2. Radiomics Model Performance and Comparison with Radiologists

Three features from the intramodular ROI (Sphericity, Correlation, and Dependence Entropy) and one feature from the perinodular ROI (Dependence Non-Uniformity Normalized) met the feature selection criteria (selected as the top 10 features more than 60 times out of 100 repetitions of LASSO fitting). Two radiomics SVM models were trained using the following features: (1) intranodular SVM model with the three radiomics features from the intranodular ROI and (2) combined SVM model with the three radiomics features from the intramodular ROI and one radiomics feature from the perinodular ROI. 

The performance of two radiomics SVM models and two radiologists is summarized in Table 3 and Figure 3. The intranodular SVM classifier achieved the sensitivity, specificity, and AUC of 0.545, 0.828, and 0.826 in the internal testing dataset, respectively. In the external testing dataset, the intranodular SVM classifier achieved the sensitivity, specificity, and AUC of 0.833, 0.964, and 0.956, respectively. The combined SVM classifier achieved the sensitivity, specificity, and AUC of 0.545, 0.966, and 0.828 in the internal testing dataset, respectively. In the external testing dataset, the combined SVM classifier achieved the sensitivity, specificity, and AUC of 0.833, 1.000, and 0.957, respectively. The expert radiologists demonstrated the diagnostic performances that were comparable to those of radiomics SVM classifiers. The radiologist reader 1 achieved the sensitivity and specificity of 0.545 and 0.966 in the internal testing group, and 0.917 and 0.929 in the external testing group, respectively. The radiologist reader 2 achieved the sensitivity and specificity of 0.636 and 0.828 in the internal testing group, and 0.833 and 0.929 in the external testing group, respectively. 

### 3.3. Associations between the Radiomics and Semantic CT Imaging Features

The correlation between the four semantic CT imaging features that showed a significant difference in conventional CT analysis (spiculated margin, subsolid density, air-bronchogram, and pleural tag) and the four radiomics features used to construct the combined SVM radiomics model (Sphericity, Correlation, and Dependence Entropy from the intranodular ROI and Dependence Non-Uniformity Normalized from the perinodular ROI) was analyzed. Sphericity, a shape radiomics feature from the intranodular ROI, showed a negative correlation with semantic CT features, spiculated margin (Spearman‘s ρ = −0.6, *p* < 0.001), subsolid density (ρ = −0.38, *p* < 0.001), air-bronchogram (ρ = −0.45, *p* < 0.001), and pleural tag (ρ = −0.48, *p* < 0.001), respectively. Correlation, a texture radiomics feature from the intranodular ROI, showed a positive correlation with semantic CT features, spiculated margin (ρ = 0.36, *p* < 0.001), subsolid density (ρ = 0.34, *p* < 0.001), air-bronchogram (ρ = 0.3, *p* < 0.001), and pleural tag (ρ = 0.26, *p* < 0.001), respectively. Dependence Entropy, a texture radiomics feature from the intranodular ROI, showed a positive correlation with semantic CT features, spiculated margin (ρ = 0.36, *p* < 0.001), subsolid density (ρ = 0.34, *p* < 0.001), air-bronchogram (ρ = 0.3, *p* < 0.001), and pleural tag (ρ = 0.26, *p* < 0.001), respectively. Dependence Non-Uniformity Normalized, a texture radiomics feature from the perinodular ROI, showed a positive correlation with semantic CT features, spiculated margin (ρ = 0.31, *p* < 0.001), and pleural tag (ρ = 0.18, *p* < 0.01), respectively. The detailed correlation matrix between the semantic CT imaging and radiomic features is shown in Figure 4.

## 4. Discussion

This study demonstrated the feasibility of using machine learning models trained with CT-based radiomics features to distinguish primary LC from solitary LM in patients with CRC. The combined SVM model, which was trained using features from both the intranodular and perinodular ROIs, demonstrated a slightly higher level of performance than the SVM model trained using features from only the intramodular ROI. In the external test set, the combined SVM classifier demonstrated an AUC of 0.957 and accuracy of 0.950, while it showed an AUC of 0.828 and accuracy of 0.850 using the internal test set. The performances of the SVM models were comparable to the performance of the radiologists (Figure 3).

Radiomics analysis provides a quantitative approach to the interpretation of medical imaging data, enabling the quantification of morphologic features such as shape, size, and volume, as well as intensity and texture features that are difficult for human readers to assess. In a previous study, Hu et al. developed a nomogram model utilizing radiomics features from the CT images of indeterminate pulmonary nodules along with clinical information in patients with CRC, and demonstrated its discriminatory ability for a metastasis prediction [17]. While this study focused on predicting the likelihood of indeterminate pulmonary nodules being metastatic in patients with CRC, our study focused on developing CT radiomics-based machine learning models for discriminating between primary LC and solitary LM in suspicious solitary pulmonary nodules with malignant potential. As primary LC and solitary LM have distinct pathogenetic, histopathologic, and immunohistochemical characteristics [18], these differences may be revealed in gross morphology and manifested as the internal heterogeneity of the nodules. A previous study showed that semantic CT imaging findings such as spiculated margin, subsolid density, and air-bronchogram sign were helpful in distinguishing primary LC and solitary LM [19]. Our results presented a consistent finding, in which these semantic CT features, including spiculated margin, subsolid density, air-bronchogram sign, and pleural tag, were significantly different between primary LC and solitary LM in the univariate analysis (Table 2). In addition, these CT features exhibited a high degree of correlation with the radiomics features that were selected to be used to train the machine learning classifiers for distinguishing the two pulmonary nodules.

In the radiomics feature selection using the LASSO method, four radiomic features, Sphericity, Correlation, Dependence Entropy, and Dependence Non-uniformity Normalized, were identified to be used for training machine learning models. Sphericity belongs to the category of shape feature, while Correlation, Dependence Entropy, and Dependence Non-uniformity Normalized belong to the category of texture features. In our study, the primary LC group had a significantly lower level of Sphericity than the solitary LM group (0.63 ± 0.08 and 0.73 ± 0.06 for primary LC and solitary LM, respectively, *p* < 0.001 by independent *t*-test). Sphericity has been shown to have a negative correlation with spiculated or irregular borders of tumors [20]. Our results are consistent with this finding, exhibiting a significant negative correlation between Sphericity and spiculated margin. The presence of spiculated margins has been shown to be a useful determinant in differentiating primary LC from pulmonary metastasis in other studies [8,19]. Correlation is one of the Gray Level Co-occurrence Matrix (GLCM) features proposed by Haralick et al. [21] and has been used as a way to texture analysis with various application in medical imaging analysis [22,23,24]. It measures the linear dependency of gray levels on the neighboring voxels and is used to evaluate the spatial relationship of an image voxel to its neighbor. GLCM Correlation has been shown to be positively correlated with heterogeneity [25]. Dependence entropy is one of the Gray Level Dependence Matrix (GLDM) features proposed by Sun et al. [26]. A large Dependence entropy indicates a high degree of complex textures in images. Dependence Non-Uniformity Normalized is another GLDM feature and implies the similarity of dependence throughout the image, with a low value indicating more homogeneity and a high value more heterogeneity in the image. In our study, the semantic CT features that were related to the heterogeneous morphology and internal characteristics, including spiculated margin, subsolid, density, air-bronchogram, and pleural tag, were more commonly observed in primary LC than in solitary LM. The primary LC group possessed significantly higher levels of Correlation, Dependence entropy, and Dependence Non-Uniformity Normalized than the solitary LM group (Appendix A). Correlation and Dependence entropy had a statistically significant positive correlation with all semantic CT features related to the heterogeneous morphology and internal characteristics. Dependence Non-Uniformity Normalized from the perinodular ROI showed a statistically significant correlation with spiculated margin and pleural tag, which are related to the heterogeneous morphology.

In our study, both the SVM models and radiologists showed high specificities and relatively low sensitivities in distinguishing primary LC from solitary LM. The number of solitary LM patients included in the training dataset (*n* = 101) was higher than that of primary LC patients (*n* = 58), and this imbalance in the training dataset may have caused the model performance with a high specificity and a relatively low sensitivity. Another reason can be due to the relatively small size of nodules included in this study. In small pulmonary nodules, morphological features are not as evident as in relatively large pulmonary nodules, making it difficult to characterize distinct imaging features between primary LC and solitary LM. Previous studies have reported that the primary LC with different sizes possessed the varying degrees of morphology, and the characteristic morphologic features of primary LC, such as spiculated margin, pleural retraction, or internal air-bronchogram, were less frequently observed in small primary LC [19,27]. In our study, the solid primary LC with a relatively small size may not have revealed the typical morphologic features and, consequently, were categorized as solitary LM by both the SVM models and radiologists.

In our study, we observed that the model performances differed between the internal and external datasets. One of the reasons for this difference may be attributed to the relatively small sample size of the testing dataset constructed with only histopathologically confirmed SPNs. In addition, we believe that the dissimilar performance between the internal and external test datasets may also originate from the inherent differences stemming from varying imaging protocols, management processes, and patient characteristics between three different institutions. As we can see from the performances of models and radiologists in Table 3, the radiologists and SVM classifiers exhibited the same performance pattern between the internal and external test sets: a low sensitivity in the internal test set and a relatively high sensitivity in the external test set. These results tell us that the external test data may have consisted of datasets that were relatively easy to distinguish. To further validate our prediction model, multiple external testing datasets that encompass larger, multi-center patient cohorts are needed [28].

Our study had several limitations. First, since our study was performed on pathologically confirmed SPN through surgical resection, most patients possessed SPNs that were localized, relatively small, and surgically resectable. The possibility of selection bias due to these conditions cannot be excluded. Second, our classifier models were built using only imaging features obtained from the radiomics analysis. In a real practice, the addition of clinical or laboratory information may be helpful in distinguishing primary LC from solitary LM. Future studies are warranted to construct machine learning models that utilize both imaging features and clinical information for an improved performance. Third, we performed the models validation with a single external test dataset with a relatively small sample size. 

## 5. Conclusions

In conclusion, we demonstrated the feasibility of using the radiomics-based machine learning classifier for distinguishing between the primary LC and solitary LM using the SPN in CRC patients. The performance of our model was comparable to those of expert radiologists. Our approach may provide an objective way to distinguish distinct pathologies of SPN in patients with CRC and open a new possibility for SPN evaluation.

## Figures and Tables

**Figure 1 jpm-12-01859-f001:**
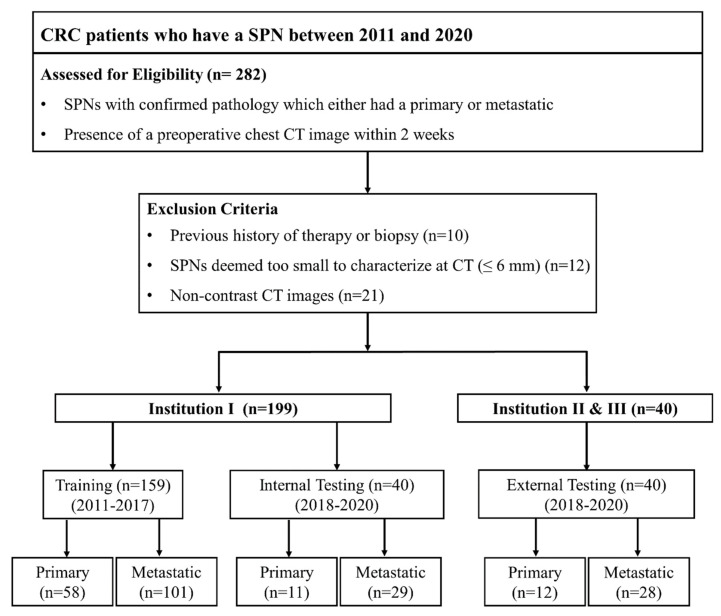
Flow diagram of the study.

**Figure 2 jpm-12-01859-f002:**
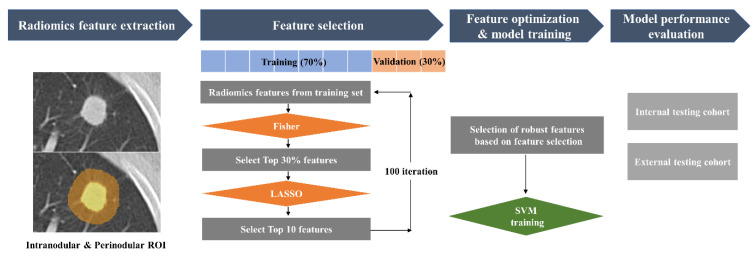
The process of the radiomic SVM model. Fisher, Fisher linear discriminant analysis; LASSO, Least Absolute Shrinkage and Selection Operator; SVM, support vector machine.

**Figure 3 jpm-12-01859-f003:**
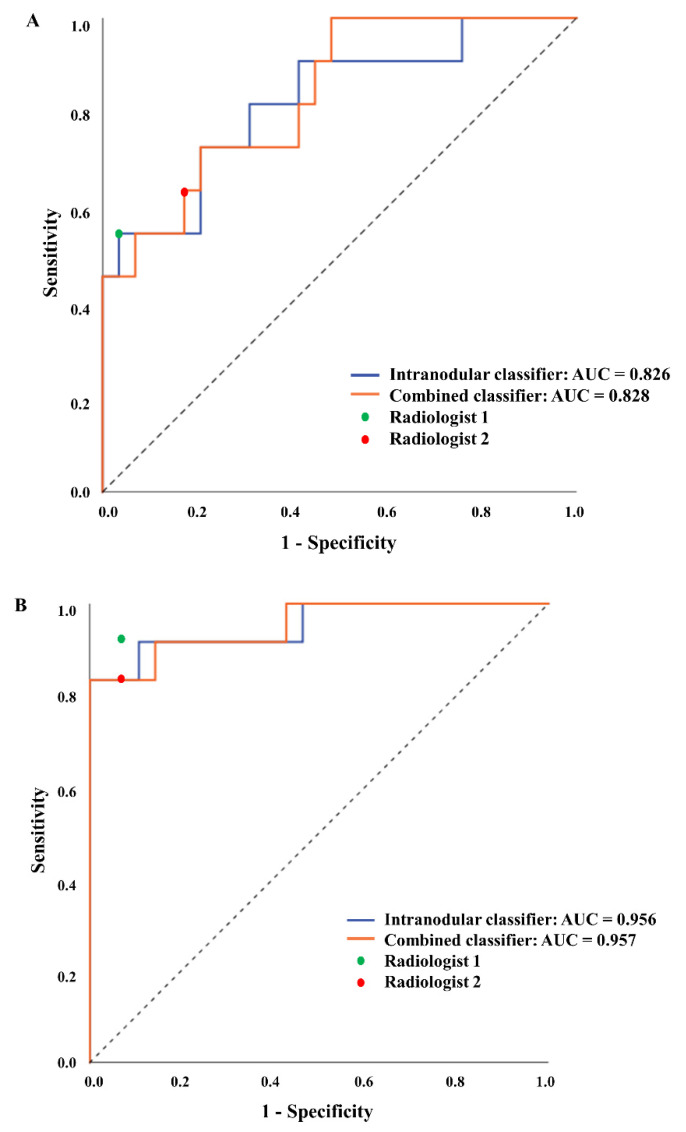
Graphs show the area under the receiver operating characteristic curve (AUC) of intranodular and combined classifiers in the internal (**A**) and external (**B**) testing datasets. The sensitivity and specificity of two radiologists are plotted as green and red dots, respectively.

**Figure 4 jpm-12-01859-f004:**
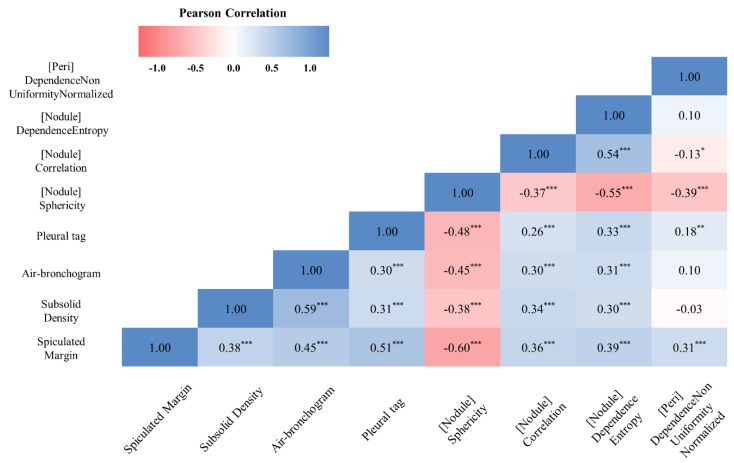
Associations between the radiomics and the categorical semantic CT imaging features. The correlations were assessed with a Pearson correlation coefficient. * Correlation is significant at the 0.05 level (2-tailed). ** Correlation is significant at the 0.01 level (2-tailed). *** Correlation is significant at the 0.001 level (2-tailed).

**Table 1 jpm-12-01859-t001:** Clinical characteristics of patients in the training, internal, and external testing datasets.

	Training (*n* = 159)	Internal Testing (*n* = 40)	External Testing (*n* = 40)
**Age (years)**	65.9 ± 9.96	65.8 ± 10.68	66.1 ± 8.37
**Sex**			
Male	107 (67.3)	24 (60.0)	26 (65.0)
Female	52 (32.7)	16 (40.0)	14 (35.0)
**History of smoking**			
Never	88 (55.3)	25 (62.5)	21 (52.5)
Yes	71 (44.7)	15 (37.5)	19 (47.5)
**Index tumor location**			
Colon	63 (39.6)	25 (62.5)	20 (50.0
Rectum	96 (60.4)	15 (37.5)	20 (50.0)
**Index tumor T stage**			
T1-2	34 (21.4)	10 (25.0)	6 (15.0)
T3-4	126 (78.6)	30 (75.0)	34 (85.0)
**Index tumor N stage**			
N0	66 (41.5)	17 (42.5)	12 (30.0)
N1-2	93 (58.5)	23 (57.5)	28 (70.0)
**Extrathoracic metastasis**			
No	139 (87.4)	37 (92.5)	38 (95.0)
Yes	20 (12.6)	3 (7.5)	2 (5.0)
**Histopathology of SPN**			
Metastatic	101 (63.5)	29 (72.5)	28 (70.0)
Primary	58 (36.5)	11 (27.5)	12 (30.0)

Note: Values in parentheses indicate percentages. Values are presented as mean ± standard deviation where applicable. SPN, solitary pulmonary nodule.

**Table 2 jpm-12-01859-t002:** Semantic CT imaging features of solitary pulmonary nodules.

	Primary LC (*n* = 81)	Solitary LM (*n* = 158)	*p* Value
**Size (mm)**	21.4 ± 7.7	14.7 ± 6.2	**<0.001**
**Margin**			**<0.001**
Smooth	9 (11.1)	78 (49.4)	
Lobulated	34 (42.0)	70 (44.3)	
Spiculated	38 (46.9)	10 (6.3)	
**Density**			**<0.001**
Solid	54 (66.7)	157 (99.4)	
Subsolid	27 (33.3)	1 (0.6)	
**Air-bronchogram**			**<0.001**
Absent	48 (59.3)	151 (95.6)	
Present	33 (40.7)	7 (4.4)	
**Cavitation**			0.459
Absent	72 (88.9)	135 (85.4)	
Present	9 (11.1)	23 (14.6)	
**Pleural tag**			**<0.001**
Absent	34 (42.0)	126 (79.7)	
Present	47 (58.0)	32 (20.3)	

Note: significant *p* values are in bold. Values in parentheses are percentages. Values are presented as mean ± standard deviation where applicable. LC, lung cancer; LM, lung metastasis.

**Table 3 jpm-12-01859-t003:** Diagnostic performance of the radiomics SVM models and radiologist readers.

	Internal Test			External Test		
	Sensitivity	Specificity	AUC	Sensitivity	Specificity	AUC
**SVM models**						
Intranodular	0.545	0.828	0.826	0.833	0.964	0.956
Combined	0.545	0.966	0.828	0.833	1.000	0.957
**Radiologists**						
Reader 1	0.545	0.966	-	0.917	0.929	-
Reader 2	0.636	0.828	-	0.833	0.929	-

Note: SVM, support vector machine; AUC, area under the receiver operating characteristic curve.

## Data Availability

The data presented in this study are available on request from the corresponding author.

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
