# Peer review of "A Radiomics Approach on Chest CT Distinguishes Primary Lung Cancer from Solitary Lung Metastasis in Colorectal Cancer Patients"

_jpm, 2022, doi:10.3390/jpm12111859_

Round 1

Reviewer 1 Report

Authors need to incorporate the following suggestions.

1. The focus on machine learning aspects (SVM) is not presented effectively. Authors need to pull interesting observations from the analysis made.

2. Cross validation could help the readers to intrepret the robustness of the proposed approach. 

3. Performance analysis section need to be strengthened. 

4. The research gaps and motivation can be consolidated at the end of the introduction section. 

5. Novelty of the proposed approach need to be emphasized properly. 

Author Response

R1-1. The focus on machine learning aspects (SVM) is not presented effectively. Authors need to pull interesting observations from the analysis made.

Response) We thank the reviewer for this comment. We added the explanation about SVM in the Feature Selection and Model Training section of Materials and Methods.

R1-2. Cross validation could help the readers to intrepret the robustness of the proposed approach. 

Response) We thank the reviewer for this comment. In our study, we implemented the bootstrapping strategy to evaluate the proposed method, which repeated the experiment 100 times with different data partitions for each iteration. Similar to the cross-validation, we believed that the bootstrapping method can enhance the robustness of the model.

R1-3. Performance analysis section need to be strengthened. 

Response) We thank the reviewer for this comment. In discussion, we explaned the potential reason for the pattern of model performance shown in the internal and external test sets. Please refere to the response to the reviewer 2 comment (R2-5) for detail.

R1-4. The research gaps and motivation can be consolidated at the end of the introduction section. 

Response) We thank the reviewer for this suggestion. We added the following sentence in the last paragraph of the introduction to consolidate the research gaps and motivation:
“To date, there have been no studies that quantitatively analyzed SPN in patients with CRC using a radiomics approach and evaluated its feasibility through comparison with radiologists.”

R1-5. Novelty of the proposed approach need to be emphasized properly. 

Response) We thank the reviwer for pointing this out. As we have explained in the response to the comment above (R1-4), one of the novelties of our study was that we analyzed SPN in patients with CRC using a quantitative radiomics approach and evaluated its feasibility through comparison with radiologists. In addition, we added the following sentence in the conclusion:

“Our approach may provide an objective way to distinguish distinct pathologies of SPN in patients with CRC and open a new possibility for SPN evaluation.”

Reviewer 2 Report

Introduction:

1.  Introduction is well written. One thing that I wondered if would be relevant to your introduction is bringing bringing data from the literature about the percentage of CRC patients who present with solitary lung nodules alone (vs. hepatic mets or other mets associated with it). Table 1 mentions about the percentage of extra-thoracic metastasis, so I wonder how that is related to the literature, just to give some context about the importance of this topic.

Methods/results:

1. I would try to be consistent in your wording. Important to distinguish and make it clear test vs. validation, especially for Figure 1 and Figure 2, I think you are using the terms interchangeably. 

2. Is there a reason why authors decided to use split sample for K-fold cross-validation with such a small dataset?

3. The parameters accuracy and AUC are meaningless clinically speaking. I understand the point of using it, but there is little clinical utility. I would recommend emphasizing sensitivity/specificity and ok to use AUC to give an "overall" performance adjusted for threshold, but reporting accuracy is trivial if you can provide more clinically relevant info such as sensitivity/specificity. Also, when reporting in the text, I would recommend using sensitivity, specificity, and AUC, in this order, as the latter is independent of the threshold and would provide the point of summarizing the overall test fit. 

4. I wonder why the authors split the sample to have one small internal test set. I think the model could have been more optimized by using the entire data for training/validation and just reporting the performance of the external test, which is what matters the most. I would recommend using the entire dataset for training/validation so optimize the model and then testing it with your external dataset. It is hard to believe, if it was not for the small sample, that your external test would provide higher diagnostic performance than internal; this way you can avoid this problem.

Discussion:

1. Please include the small dataset in your limitations. Also, as above, the differences in your ext vs. internal could be related to that.

2. You could make the point of having a small dataset because only indeterminate nodules are being biopsied and therefore it is not easy to gather these patients. 

Author Response

R2-1.  Introduction is well written. One thing that I wondered if would be relevant to your introduction is bringing bringing data from the literature about the percentage of CRC patients who present with solitary lung nodules alone (vs. hepatic mets or other mets associated with it). Table 1 mentions about the percentage of extra-thoracic metastasis, so I wonder how that is related to the literature, just to give some context about the importance of this topic.

Response) Thank you for your advice. Since this study only targeted CRC patients with surgically resected SPN, it was difficult to compare the accompanying extrathoracic metastasis rate with other literatures. However, we added the following sentence with the relevant reference in the introduction to provide more context to our study :

“Solitary pulmonary nodule (SPN) is often encountered in patients with colorectal cancer (CRC). In a previous study of pulmonary nodules in CRC patients, 27% of all patients had pulmonary nodules, of which 56% were SPN.”

Methods/results:

R2-2. I would try to be consistent in your wording. Important to distinguish and make it clear test vs. validation, especially for Figure 1 and Figure 2, I think you are using the terms interchangeably. 

Response) We thank the reviwer for this comment and apologize for the confusion. We revised the manuscript thoroughly, including Figure 1 and Figure 2, to make terms consistent. We divided the entire data into training, internal testing, and external testing.

R2-3. Is there a reason why authors decided to use split sample for K-fold cross-validation with such a small dataset?

Response) In our study, instead of K-fold validation, we implemented the bootstrapping strategy to evaluate the proposed method, which repeated the experiment 100 times with different, random data partitions for each iteration. We believed that the bootstrapping method can enhance the robustness of the model for the feature selection step, especially in a model with a small dataset. For example, the radiomics feature, Sphericity, was always selected as one of the top-10 features by LASSO over 100 iterations and we were confident that Sphericity had a strong contribution to distinguish between LC and LM.

R2-4. The parameters accuracy and AUC are meaningless clinically speaking. I understand the point of using it, but there is little clinical utility. I would recommend emphasizing sensitivity/specificity and ok to use AUC to give an "overall" performance adjusted for threshold, but reporting accuracy is trivial if you can provide more clinically relevant info such as sensitivity/specificity. Also, when reporting in the text, I would recommend using sensitivity, specificity, and AUC, in this order, as the latter is independent of the threshold and would provide the point of summarizing the overall test fit. 

Response) We thank the reviewer for this suggestion. We revised the results to show sensitivity, specificity and AUC in order as you have recommended.

R2-5. I wonder why the authors split the sample to have one small internal test set. I think the model could have been more optimized by using the entire data for training/validation and just reporting the performance of the external test, which is what matters the most. I would recommend using the entire dataset for training/validation so optimize the model and then testing it with your external dataset. It is hard to believe, if it was not for the small sample, that your external test would provide higher diagnostic performance than internal; this way you can avoid this problem.

Response) Thank you for your valuable advice. In our study, the dataset came from 3 different hospitals: Chonnam National University Hwasun Hospital (n=199), Chonnam National University Hospital (n=20), and Jeonbuk National University Hospital (n=20). Most of the data came from the 1st hospital and we used 20% of the 1st hospital data for internal test. We agree with the reviwer that the small internal test set may have caused the low performance in the internal test. In addition, we believe that the dissimilar performance between the internal and external test sets may originate from the inherent difference in patient characteristics between different institutions. Even though we applied the same inclusion and exclusion criteria for patient recruitment, there may exist inherent differences stemming from varying imaging protocols, management processes, and patient characteristics between different institution. As we can see from the performances of models and radiologists in Table 3, the radiologists and SVM classifiers exhibited the same performance pattern between the internal and external test sets: a low sensitivity in the internal test set and a relatively high sensitivity in the external test set. These results tell us that the external test data may have consisted of dataset that were relatively easy to distinguish. Multiple external testing datasets that encompass a larger, multi-center patient cohorts are needed to further validate our prediction model. For an improved analysis of results including the above-menthioned points, we added the following paragraph in the discussion:

“In our study, we observed that the model performances differed between the internal and external datasets. One of the reasons for this difference may be attributed to the relatively small sample size of the testing dataset constructed with only histopathologically confirmed SPNs. In addition, we believe that the dissimilar performance between the internal and external test datasets may also originate from the inherent differences stemming from varying imaging protocols, management processes, and patient characteristics between three different institutions. As we can see from the performances of models and radiologists in Table 3, the radiologists and SVM classifiers exhibited the same performance pattern between the internal and external test sets: a low sensitivity in the internal test set and a relatively high sensitivity in the external test set. These results tell us that the external test data may have consisted of dataset that were relatively easy to distinguish. To further validate our prediction model, multiple external testing datasets that encompass a larger, multi-center patient cohorts are needed”

Additionaly, we followed the reviwer’s suggestion and re-trained and optimized our models using the entire dataset from the 1st hospital (train + internal test sets in the original manuscript) and tested the model performance using the external test set. Unfortunatley, the new results demostratred a similar pattern: a low sensitivity in validation (20% of the entire 1st hospital data) and a relatively high sensitivity in the test set.

Discussion:

R2-6. Please include the small dataset in your limitations. Also, as above, the differences in your ext vs. internal could be related to that.

Response) Thank you for this comment. We added the discussion as mentioned above and revised the limitation as follow:

“Third, we performed the models validation with a single external test dataset with relatively small sample size.”

R2-7. You could make the point of having a small dataset because only indeterminate nodules are being biopsied and therefore it is not easy to gather these patients.

Response) Thank you for pointing this out. We revised the limitation of the manuscript as mentioned above.

Round 2

Reviewer 1 Report

The paper organization is not standardized. Contents need to be fitted properly inside each section with proper section numbers and names. 

Author Response

The paper organization is not standardized. Contents need to be fitted properly inside each section with proper section numbers and names.

Response) We apologize for not understanding the standardized instruction, thoroughly. We reorganized the manuscript using proper section numbers as you commented.

Reviewer 2 Report

I think the Authors were thoughtful in replying to my concerns and although the paper has many limitations, it may contribute to the current literature.

Author Response

I think the Authors were thoughtful in replying to my concerns and although the paper has many limitations, it may contribute to the current literature.

Response) We thank for your thoughtful and valuable comments.